# Gemcitabine: An Alternative Treatment for Oxaliplatin-Resistant Colorectal Cancer

**DOI:** 10.3390/cancers14235894

**Published:** 2022-11-29

**Authors:** Mathieu Chocry, Ludovic Leloup, Fabrice Parat, Mélissa Messé, Alessandra Pagano, Hervé Kovacic

**Affiliations:** 1Institut de Neurophysiopathologie (INP, UMR 7051), CNRS, Faculté de Médecine, Aix-Marseille University, 13385 Marseille, France; 2Laboratoire de Bioimagerie et Pathologies (LBP), UMR CNRS 7021, Faculté de Pharmacie, Université de Strasbourg, 67401 Illkirch, France

**Keywords:** colorectal cancer, chemoresistance, oxaliplatin, gemcitabine, p38 MAPK

## Abstract

**Simple Summary:**

Colorectal cancer is the third most common cancer worldwide. The treatment of the advanced stages is based on poly-chemotherapies, including oxaliplatin. However, the development of resistance to chemotherapy is observed in 50% of cases, leading to treatment failures. A better understanding of the resistance mechanisms is therefore crucial to improve treatment efficiency and patient survival. In our previous work, showed that ROS production and the p38 MAPK pathway were strongly involved in resistance to oxaliplatin. In this study, we tested several chemotherapies and observed that only gemcitabine efficiently treated oxaliplatin-resistant cancer cells. Indeed, gemcitabine was able to induce apoptosis by inhibiting both the Akt and p38 MAPK pathways. Taken together, our results show that gemcitabine could be an interesting therapeutic option for patients with oxaliplatin-resistant tumors.

**Abstract:**

Resistance to treatments is one of the leading causes of cancer therapy failure. Oxaliplatin is a standard chemotherapy used to treat metastatic colorectal cancer. However, its efficacy is greatly reduced by the development of resistances. In a previous study, we deciphered the mechanisms leading to oxaliplatin resistance and highlighted the roles played by ROS production and the p38 MAPK pathway in this phenomenon. In this report, we studied the effects of different chemotherapy molecules on our oxaliplatin-resistant cells to identify alternative treatments. Among all the studied molecules, gemcitabine was the only one to present a major cytotoxic effect on oxaliplatin-resistant cancer cells both in vivo and in vitro. However, the combination of oxaliplatin and gemcitabine did not present any major interest. Indeed, the study of combination efficiency using Chou and Talalay’s method showed no synergy between oxaliplatin and gemcitabine. Using PamGene technology to decipher gemcitabine’s effects on oxaliplatin-resistant cells, we were able to show that gemcitabine counteracts chemoresistance by strongly inhibiting the Akt and src/p38 MAPK pathways, leading to apoptosis induction and cell death. In view of these results, gemcitabine could be an interesting alternative therapy for patients with colorectal cancer not responding to oxaliplatin-based protocols such as FOLFOX.

## 1. Introduction

Colorectal cancer is the fourth most frequent cancer and the third most frequent cause of cancer death worldwide [1]. Its mortality rate is one of the highest, especially due to the therapeutic escape observed for advanced stages. Despite the recent progress made in terms of combination chemotherapies, the five-year survival rate for metastatic colorectal cancer remains very low: 13.1% in the USA and less than 10% in Europe. 

The treatment of metastatic colorectal cancer is based on poly-chemotherapies. These combinations are usually composed of 5-fluoro-uracil, folinic acid, oxaliplatin, and/or irinotecan [2]. These poly-chemotherapies can be coupled to monoclonal antibodies such as Cetuximab. However, the effectiveness of these monoclonal antibodies depends on the mutational status of the tumor cells. For example, Cetuximab has been shown to be ineffective in patients whose tumors have K-Ras or B-Raf mutations [3,4]. The treatment failure is not only due to the mutational status of the patient but also, in around 50% of cases, to the development of resistance to chemotherapies [5]. It is essential to identify resistance mechanisms in order to propose alternative treatments that can improve treatment efficiency.

There are many resistance mechanisms to chemotherapy. Some are specific to each molecule, and others are common to all chemotherapies or to groups of related anticancer drugs. The modification of drug availability by the alteration of influx or efflux via MRP is one of the most well-known multidrug resistance mechanisms [6,7]. There are also specific mechanisms to resist the apoptotic process induced by chemotherapies. For example, an increase in the expression of two apoptosis inhibitors, survivin and BIRC6, leads to decreased sensitivity of colorectal cancer cells to oxaliplatin [8,9]. In our previous study, we described a new pathway responsible for the resistance of CRC cells to oxaliplatin. We notably showed that this pathway leads to the inhibition of apoptosis through an abnormal activation of p38 MAPK [10]. These kinases, and in particular p38 α, are well known for their anti-apoptotic activity and were shown to be interesting therapeutic targets in colorectal cancer [11,12]. However, p38 α was also shown to exert pro-apoptotic effects in normal cells [13]. Targeting p38 MAPK is therefore complicated because of the wide spectrum of action of their inhibitors [14]. 

In this work, we studied different alternatives to treat oxaliplatin-resistant cancer cells, and we demonstrate that a chemotherapy, gemcitabine, could be used to overcome resistance to oxaliplatin. This chemotherapy is the only one able to induce apoptosis in our resistant cells, via an inhibition of the Akt and p38 MAPK pathways. We also show that the combination of oxaliplatin and gemcitabine is not a good alternative to poly-chemotherapy since this combination is antagonist in all our resistant cells. Taken together, our data show that a monotherapy of gemcitabine could be a good alternative for the treatment of patients with colorectal cancer that is not responding to oxaliplatin.

## 2. Materials and Methods

### 2.1. Tumor Cell Lines and Culture Conditions

The human colorectal cancer cell lines used in this study (HT29-D4, Caco-2, and RKO) were growth in DMEM medium (Dulbecco’s modified Eagle’s medium, Thermo Fisher Scientific, Waltham, MA, USA) supplemented with 10% FBS (fetal bovine serum) and L-glutamine (2 mM). Non-essential amino-acids were also added to the medium used to maintain Caco-2 cells. The cells were incubated at 37 °C in a humidified atmosphere with 5% CO_2_. The HT29-D4 cells were originally selected from HT29 colon adenocarcinoma cells by Fantini et al. [15]. The oxaliplatin-resistant cells (Rox) were selected as described in our previous work [10]. These cells were cultured in the same medium as their parental cells, supplemented with 2 μM oxaliplatin (LOHP, Sigma-Aldrich, Merck, Darmstadt, Germany).

### 2.2. Reagents and Antibodies

The following reagents were used: oxaliplatin (LOHP, Sigma-Aldrich, Merck, Darmstadt, Germany) stored at 5.4 mg/mL (12.5 mM); gemcitabine (gemcitabine hydrochloride, Sigma-Aldrich, Merck, Darmstadt, Germany) stored at 30 mg/mL (100 mM); doxorubicin (doxorubicin hydrochloride solution, Sigma-Aldrich, Merck, Darmstadt, Germany) stored at 2 mg/mL (3.5 mM); 5-fluoro-uracil (5-fluorouracil, Sigma-Aldrich, Merck, Darmstadt, Germany) stored 130 mg/mL (1 M); etoposide (Sigma-Aldrich, Merck, Darmstadt, Germany) stored at 25 mg/mL (42.4 mM); vincristine (vincristine sulfate salt, Sigma-Aldrich, Merck, Darmstadt, Germany) stored at 1 mg/mL (1 mM); and paclitaxel (Sigma-Aldrich, Merck, Darmstadt, Germany) stored at 1mg/mL (1.2 mM); all these chemotherapies were diluted in PBS (PBS + 5% glucose for oxaliplatin) and used at different concentrations. The following primary antibodies were used: anti-Akt (diluted at 1/1 000, ref. 9272, Cell Signaling Technology, Danvers, MA, USA); anti-phospho-Akt (Ser473) (diluted at 1/1 000, ref. 9271, Cell Signaling Technology, Danvers, MA, USA); anti-GAPDH (1/20 000, ref. G8795, Sigma-Aldrich, Merck, Darmstadt, Germany); anti-p38 (1/1 000, ref. sc-535, Santa Cruz Biotechnology, Santa Cruz, CA, USA); anti-phospho-p38 (Thr180/Tyr182) (1/500, ref. sc17852, Santa Cruz Biotechnology, Santa Cruz, CA, USA); anti-p70S6K (1/1 000, ref. 9202, Cell Signaling Technology, Danvers, MA, USA); and anti-phospho-p70S6K (Thr 389) (1/1 000, ref. 9205, Cell Signaling Technology, Danvers, MA, USA). The HRP-coupled secondary antibodies were purchased from Cell Signaling Technology.

### 2.3. Cytotoxicity Assay

Cell viability was measured using MTT assays. The sensitive or resistant (Rox) cells were counted and plated in 96-well plates (50,000 cells/mL). After 24 hours, the cells were treated with increasing concentrations of the chemotherapy agents (from 0 to 100 µM or nM, according to the drug used). PBS (PBS + 5% glucose for oxaliplatin) was used as a control. After 72 hours of incubation, the medium was replaced with DMEM containing 0.5 mg/mL MTT, and the cells were incubated for 2 hours. The cells were then lysed, and the formazan was solubilized using pure DMSO (Sigma-Aldrich, Merck, Darmstadt, Germany). The optical density (OD) was measured using a plate reader (600 nm, Multiskan RC, Thermo-Labsystems, Waltham, MA, USA). Cell viability was expressed as a percentage of survival, using 100% for the untreated cells. The IC_50_ values were calculated by using the linearization described by Chou and Talalay [16].

### 2.4. In Vivo Experiments

All procedures and care for animals been complied with the official directives of the French government. The in vivo experiments were approved by the Regional Committee for Ethics on Animal Experiments (Marseille number: 14, authorization number: 2017073110394582###) and were performed in the Animal Facility of the Faculty of Pharmacy of Aix-Marseille University (agreement number: D 13-055-20). Female athymic nude mice (6 to 8 weeks old) were obtained from Charles Rivers Laboratories (France). One million HT29-Rox1 cells were inoculated subcutaneously into the right dorsal flank of the nude mice. Experimental treatments were started when the tumors reached about 100 mm^3^ in volume (day 0). Two groups of mice were treated with intravenous weekly injections (i.v.) of either oxaliplatin (6.7 mg/kg in PBS and 5% glucose) or only PBS and 5% glucose (*n* = 7 mice each group), according to previously described protocols [17,18]. Two groups of mice were treated every 3 days with intraperitoneal injections (i.p.) of either gemcitabine (80 mg/kg in PBS) or only PBS (*n* = 9 mice) [19].

During the experiment, twice a week, the mice were weighed, and the tumor volumes were determined by caliper measurements according to the formula (π/6) × (L × l^2^). According to the rules of the ethics committee, the mice were sacrificed when their tumor volumes reached 1500 mm^3^. Survival medians were analyzed using a Kaplan–Meier estimator. The log-rank test was used to compare survival rates in a univariate analysis. 

### 2.5. Combination Index Analysis

After counting and plating the sensitive or resistant (Rox) cells (5 × 10^4^ cells/mL) in 96-well plates with DMEM culture medium, the cells were exposed to oxaliplatin and gemcitabine, either alone or in combination, according to the described protocol [20]. The synergism of the drugs was determined using CompuSyn software (version 1.0, Ting Chao Chou and Nick Martin, Paramus, NJ, USA, 2005). The combination index (CI) values were calculated according to the Chou and Talalay method [21]. The CI values were generated using the following formula: CI = (D)1/(Dx)1+(D)2/(Dx)2, with (Dx)1 and (Dx)2 representing the doses of drugs 1 and 2 required to obtain the same effects in a combination as the single drugs 1 and 2 at D1 or D2, respectively. The combination index and the dose effect curves were generated by CompuSyn software (ComboSyn, New York, NY, USA).

### 2.6. Annexin V-FITC Assay for Apoptosis

The sensitive and resistant (Rox) cells were seeded into 6-well plates at a density of 5 × 10^5^ cells per well. After 24 hours of incubation, the cells were treated with oxaliplatin or gemcitabine for 48 hours. After treatment, cells were harvested by trypsinization and stained with Annexin V/FITC and propidium iodide (PI) in binding buffer, according to the manufacturer’s protocol (Thermo Fisher Scientific, Cat No V13242). Cells were analyzed using a flow cytometer (Gallios, Beckman Coulter, Brea, CA, USA).

### 2.7. Western Blotting

After cell lysis, 30 µg of proteins of each sample were loaded on 10% SDS-polyacrylamide gels. After separation, the proteins were transferred on a nitrocellulose blotting membrane (Amersham Protan, GE Healthcare, Chicago, IL, USA) using a Bio-Rad transfer system. The membranes were then saturated for one hour using a blocking solution (TBST plus 5% nonfat milk) and incubated overnight with the primary antibodies (in a TBST milk solution). After the incubation, the membranes were washed three times with TBST (TBS plus 0.05% Tween20) and incubated for one hour with the corresponding secondary antibodies (HRP-conjugated). The membranes were washed three times with TBST and incubated for one minute with a Merck chemiluminescence HRP substrate. The bands were revealed using Syngene G-box, and their intensities were quantified using ImageJ software (NIH) (version 1.53). For the studies of phosphorylated proteins, the results were expressed as a ratio (phosphorylated protein/total protein; 100% for the control condition). Similarly, for the study of PARP cleavage, the results were expressed as a ratio (cleaved PARP/total PARP; 100% for the control condition). All the whole western blot figures can be found in the Appendix A.

### 2.8. Kinase Activity Assay 

The kinase activities were quantified using PamGene PTK (protein tyrosine kinase) and STK (serine–threonine kinase) kits. The sensitive and resistant HT29-D4 cells were treated with 2 nM gemcitabine. After a 4-hour incubation, the cells were collected and lysed using m-PER lysis buffer (Thermo Fisher Scientific, Waltham, MA, USA). The extracted proteins were quantified using Bio-Rad Protein Assay Dye Reagent Concentrate. The samples were then loaded on PamGene PTK and STK PamChips (5 µg of proteins for PTK and 1 µg for STK). Peptide phosphorylation was monitored using the PamStation® 12, following the protocols provided by PamGene (the Netherlands). The results were analyzed and quantified using PamGene BioNavigator software (version 2.3). The fact that several peptides used in the assay are linked to a same kinase allows the BioNavigator software to calculate a specificity score using six different databases.

### 2.9. Statistical Analysis

All data were recorded in triplicate and were repeated at least three times. All in vitro test results are expressed as the means. The statistical analysis was performed with unpaired Student’s *t*-tests. A difference was considered significant when the *p* value was less than 0.05. For the kinomic analysis, image analysis and signal quantification were performed using the BioNavigator^®^ software (PamGene, ‘s-Hertogenbosch, The Netherlands). The results were analyzed using unpaired Student’s *t*-tests (resistant cells used as controls). Kinexus Kinase Predictor was used to determine the putative modified kinases. For the in vivo studies, statistical significance was assessed via a one-way ANOVA. For the Kaplan–Meyer curves, the log-rank (Mantel–Cox) test was used. In our figures, the values represent the means plus or minus the standard deviations, and significance is represented by an asterisk (*). 

## 3. Results

### 3.1. Oxaliplatin-Resistant Cells Are Sensitive to Gemcitabine

To know if the oxaliplatin-resistant cells (Rox) selected in our previous study [10] are also resistant to other drugs, we performed cytotoxic assays with the following chemotherapy molecules: 5-fluoro-uracil (5-FU), etoposide, vincristine, paclitaxel, doxorubicin, and gemcitabine. We chose these drugs because of their potential to regulate p38 MAPK. Several studies have shown that the activation of p38 MAPK can lead to the development of resistance, but on the other hand this activation was also shown to induce apoptosis [22,23,24,25,26,27,28,29,30] (Appendix A). The IC_50_ values obtained for the different chemotherapies, except for gemcitabine, are presented in Table 1. These results clearly show that the resistance to oxaliplatin also strongly reduces the sensitivity of the cells to other chemotherapies, with the notable exception of gemcitabine (results presented in Figure 1). 

Gemcitabine is a chemotherapy agent with clinical activity against a number of solid tumors, but it is not used for the treatment of colorectal cancer [31,32]. The IC_50_ values of gemcitabine were significantly decreased from 7.6 ± 1.6 nM for HT29-D4 to 1.9 ± 0.6 nM and 1.9 ± 0.5 nM for HT29-D4 Rox1 and Rox2 cells, respectively (*p* < 0.05, Figure 1A), whereas the IC_50_ values of oxaliplatin were significantly increased from 0.8 ± 0.2 µM for HT29-D4 to 5.2 ± 0.6 µM and 6.3 ± 0.9 µM for HT29-D4 Rox1 and Rox2 cells, respectively (*p* < 0.05, Figure 1B). Similarly, the IC_50_ values of gemcitabine were significantly decreased from 12.5 ± 1.1 nM for RKO to 3.2 ± 0.8 nM and 1.9 ± 0.6 nM for Rox cells, and the IC_50_ values of oxaliplatin were significantly increased from 0.5 ± 0.1 µM for RKO-sensitive cells to 13.3 ± 2.2 µM and 4.4 ± 0.2 µM for RKO Rox1 and Rox2 cells, respectively (*p* < 0.05, Figure 1C,D). 

Very similar results were obtained with Caco-2 and Caco-2-Rox cells treated with gemcitabine, with IC_50_ values of 11.6 ± 2.0 nM for sensitive cells and 2.4 ± 0.5 nM for the resistant ones (*p* < 0.05, Figure 1E), while for oxaliplatin the IC_50_ values were 0.9 ± 0.2 µM for the sensitive cells and 23.1 ± 1.4 µM for the resistant ones (*p* < 0.05, Figure 1F).

### 3.2. The Gemcitabine/Oxaliplatin Combination Presents no Interest for the Treatment of Oxaliplatin-Resistant Cells

As gemcitabine is able to reduce the viability of oxaliplatin-resistant cells, we decided to study the effects of a combination of the two molecules. Our results, presented in Figure 1D, first confirm that the HT29-D4 cells are sensitive to oxaliplatin, while this drug has no effect on Rox1 and Rox2 cell viability, as previously shown [10]. Gemcitabine alone had no significant effect on HT29-D4 cells, as their viability was only decreased to 77.4 ± 2.5%. The combination of the two chemotherapies did not bring any additional cytotoxicity compared to the oxaliplatin treatment alone. Indeed, the incubation of the two drugs together for 72 hours significantly decreased the cell viability to 46.3 ± 1.4%. This value was not significantly different from the viability obtained for oxaliplatin alone. Opposite effects were observed in oxaliplatin-resistant cells. In Rox1 cells, cell viability was significantly reduced by gemcitabine and by the combination treatment, decreasing to 25.3 ± 3.8% and 30.7 ± 1.1%, respectively. There was no significant modification between these two treatment conditions. The incubation with oxaliplatin induced no change in cell viability in the oxaliplatin-resistant cells, the viability value being 95.7 ± 3.9%. Similarly, the viability of Rox2 cells was not affected by oxaliplatin, while it was significantly reduced to 38.2 ± 3.4% and 35.9 ± 1.6% by gemcitabine and the combination treatment, respectively (Figure 1G).

As no difference was observed between the drug combination and the chemotherapy used alone, we studied the drug interaction relationship using CompuSyn software. We calculated the combination index (CI). A CI < 1 indicates a synergism, a CI > 1 indicates an antagonism, and a CI = 1 indicates an additive effect. As for the affected fractions, we observed differences between HT29-D4 cells and oxaliplatin-resistant cells. For HT29-D4 cells, the CI was under 1 for all doses except for the highest. Indeed, for 4xIC_50_, the CI was around 2.5 (Figure 1H). For the resistant cells, the CI values were close to 1 for the lowest doses and were higher for two and four times the IC_50_ (Figure 1H). The same data were obtained with the RKO and Caco-2 cells.

### 3.3. Gemcitabine Overcomes Oxaliplatin Resistance In Vivo

To validate our promising results obtained in cellulo, we studied the effects of gemcitabine on mice bearing human oxaliplatin-resistant xenografts. As shown in Figure 2A, the treatments with only PBS solutions had no effect on tumor progression, as shown by the increase in tumor volume over time. As expected, oxaliplatin was unable to inhibit tumor growth. Indeed, the average tumor size reached 1108.12 ± 135.36 mm^3^ after 31 days of treatment with oxaliplatin (Figure 2A). This value was not significantly different from the average tumor size measured for the mice treated with PBS and glucose (1075.12 ± 198.76 mm^3^).

Conversely, gemcitabine was not only able to stop tumor progression but also to significantly reduce the tumor size. After 31 days of treatment with gemcitabine, the average size of the tumors was 82.47 ± 11.48 mm^3^, while it reached 1302.17 ± 234.08 mm^3^ after the PBS treatment (Figure 2C). Gemcitabine was thus able to reduce the tumor size by 52.41%. The differences that we observed were significant after 7 days of treatment for gemcitabine and PBS and after 10 days for gemcitabine and oxaliplatin. As shown in Figure 2B, the time to endpoint of the mice treated with oxaliplatin was not significantly different from the PBS-treated mice, whereas the treatment with gemcitabine resulted in a significant increase in mouse survival (no death at the endpoint of the study) (Figure 2D).

These data were corroborated by the sizes of the resected tumors. Indeed, tumors from gemcitabine-treated mice were significantly smaller and lighter than those from the control mice, while the treatment with oxaliplatin had no effect (Figure 2E,F).

### 3.4. Gemcitabine Induces Apoptosis in Oxaliplatin-Resistant Cells

To study the effects of gemcitabine on the induction of apoptosis, we carried out annexin-V/PI double staining assays. Representative results are shown in Figure 3A–C, and the statistical analyses are presented in Figure 3D,E for early and late apoptosis + necrosis, respectively.

As expected, oxaliplatin was able to induce both early and late apoptosis in the HT29-D4 cells, while it had no effect on Rox1 and Rox2 cells. Indeed, for HT29-D4 cells the percentages of early and late apoptotic cells increased from 5.6% and 4.9% to 17.6% and 9.7% after a 48-hour incubation with oxaliplatin, respectively. For the resistant cells, no significant modification in the early or late stage of apoptosis was observed, as the values only increased from 2.6% and 1.8% to 3.8% and 3.9% for Rox1 and from 1.9% and 2.2% to 3.2% and 3.9% for Rox2.

Gemcitabine induced no significant change in early apoptosis for sensitive and resistant cells (Figure 3D). However, a significant induction of late apoptosis was observed, particularly for Rox1 and Rox2 cells. Indeed, the percentage of late apoptotic cells increased from 1.8% and 2.2% for Rox1 and Rox2, respectively, to 16.6% and 13.5% after incubation with gemcitabine (Figure 3E). To confirm our data, we studied PARP-1 cleavage in the presence of oxaliplatin or gemcitabine. As shown in Figure 3F, the oxaliplatin treatment induced the cleavage of PARP-1 only in the HT29-D4 cells, with a significant increase in the cleaved PARP/total PARP ratio from 100% to 348 ± 24%. Conversely, oxaliplatin had no effect on Rox1 and Rox2 cells, with the ratio increasing from 100% to 111 ± 27% for Rox1 and decreasing from 100% to 80 ± 73% for Rox2. Contrary to oxaliplatin, gemcitabine was unable to induce PARP cleavage in HT29-D4 cells, with a nonsignificant ratio of 158 ± 48%, while this cleavage was observed in the oxaliplatin-resistant cells with significant ratios of 2642 ± 140% and 1427 ± 418%, respectively.

### 3.5. Gemcitabine Targets Src, p38 MAP Kinase, and Akt Signaling Pathways

To identify the mechanisms used by gemcitabine to induce apoptosis in these oxaliplatin-resistant cells, we performed a screening of kinase activities using PamGene technology. The results obtained for the tyrosine kinase assay (PTK) analysis and presented in Figure 4A clearly show that Src kinase activity was decreased by the gemcitabine treatment in resistant cells in comparison to sensitive cells. Indeed, the kinases of the Src family were on the top kinase list, with high specificity scores (1.2 for SRC) and high normalized kinase statistics (−1.0 for SRC). These data show that Src kinases are strongly inhibited by gemcitabine in resistant cells. 

Concerning the serine/threonine kinase (STK) screening, the results show that the treatment of our cells with gemcitabine induced a major and significant decrease in p38 MAPK activity, as shown in Figure 4B. Indeed, the kinases of the p38 MAPK family were on the top list, with high specificity scores (2.2 for p38α MAPK (MAPK 14), 1.7 for p38γ MAPK (MAPK12), and 2.2 for p38δ MAPK (MAPK 11)) and high normalized kinase statistics (-1.6 for all of them; Figure 4B). Moreover, the STK analysis also clearly showed that the activity of the mTor/Akt/p70S6 kinase pathway was decreased by gemcitabine in the resistant cells in comparison to the HT29-D4 cells (Figure 4B). The results obtained with the PamGene technology were confirmed using Western blots (see Appendix A for p70S6 kinase and Appendix A for p38 MAPK).

To visualize the gemcitabine response pathway, the protein–protein interactions among the different PTK or STK proteins were confronted using a string functional association network (Figure 4C). Three of them, the MAPK signaling pathway, the PI3K-Akt signaling pathway, and EGFR tyrosine inhibitor resistance, with log ratios of 44.5, 26.7, and 25, respectively, were selected and analyzed (Figure 4D–F). These analyses showed a strong implication of the MAPK and PI3K-Akt signaling pathways in the responses of resistant cells to gemcitabine.

Moreover, we showed an EGFR tyrosine kinase inhibitor resistance pathway reduction in resistant cells treated with gemcitabine. All the kinases represented in the three interactomes, the MAPK signaling pathway, the PI3K-Akt signaling pathway, and EGFR tyrosine inhibitor resistance, are listed and classified in Appendix A, respectively. These data show that a combination of gemcitabine and EGFR inhibitors could be an interesting alternative for the treatment of colorectal cancers that are resistant to the FOLFOX protocol.

## 4. Discussion

In this study, we tested several chemotherapy drugs that could replace oxaliplatin for the treatment of resistant colorectal cancer cells. Our results show that oxaliplatin-resistant cells are also less sensitive to most of the drugs that we tested, with IC_50_ values 3 to 6 times higher than those of HT29-D4 cells. These results could suggest a multidrug resistance mechanism, potentially based on chemotherapy release transporters such as MRP2 (ABCC2), which was recently linked to oxaliplatin resistance [33]. Interestingly, oxaliplatin-resistant cells presented an increased sensitivity to gemcitabine, with IC_50_ values five times lower than for HT29-D4 cells. Moreover, gemcitabine treatment strongly inhibited Rox tumor progression in mice. Several studies have been carried out to test the efficiency of gemcitabine in treating colorectal cancer [34,35,36,37,38]. However, none of them showed that gemcitabine could be more effective than other chemotherapies [37]. A positive correlation was even observed between oxaliplatin and gemcitabine, showing that CRC cells that are less sensitive to oxaliplatin also tend to be less sensitive to gemcitabine (Genomic of Drug Sensitivity in Cancer, www.cancerrxgene.org, accessed on 15 November 2022). The combination of gemcitabine and oxaliplatin (GEMOX) was also tested for the treatment of colorectal cancers. However, the results showed that this combination is not efficient [17]. Therefore, the combination of gemcitabine and oxaliplatin has no therapeutic interest to treat colorectal cancer, and gemcitabine should be used alone. 

The results that we present in this study were mainly obtained using the HT29-D4 cell line. These cells, developed in our laboratory [15], present the BRAF V600E mutation. This mutation is highly described in colorectal cancers, particularly since it is associated with a low 5-year survival rate [39]. However, we also used other cell lines (RKO and Caco-2) that present different mutations, and we obtained very similar results. We can therefore infer that gemcitabine could be used regardless of the mutation status of the patients.

To have a better understanding of the mechanism of action of gemcitabine, we studied the induction of apoptosis. Several studies have shown that different platinum salts, including oxaliplatin, induce apoptosis via caspase activation [40]. In our previous work, we clearly showed that oxaliplatin-resistant cells (Rox) escaped the treatment through an inhibition of apoptosis. Here, we showed that gemcitabine is able to induce apoptosis in oxaliplatin-resistant cells, as shown by the increase in markers of late apoptosis and of PARP-1 cleavage. This observation confirms the fact that the resistance to oxaliplatin that we described in our previous work is due to an inhibition of the apoptotic pathways.

We had previously shown that p38 MAPK was responsible for this inhibition. We therefore studied the effects of gemcitabine on the kinome. Our data showed that gemcitabine inhibits both Src kinases and p38 MAPK. By inhibiting Src, gemcitabine could reduce the calpain-dependent Nox1 activity, thus decreasing the production of ROS and the activation of p38 MAPK. The inhibition of this kinase by gemcitabine may be responsible for the restoration of apoptosis. Our results also showed an inhibition of the Akt/mTor/p70S6 signaling pathway by gemcitabine. The Akt signaling pathway is well known to induce cell survival, and it is not surprising that gemcitabine inhibits this pathway. Moreover, the activation of Akt has already been shown to be involved in the resistance of tumor cells to gemcitabine [41,42,43]. A preclinical study was carried out with gemcitabine supplemented with an Akt inhibitor, LY2780301, to improve the efficiency of the chemotherapy [44]. This study was performed on several types of solid tumors. However, it only included two patients with colorectal cancer. Therefore, these data are not sufficient to be conclusive [44]. As several mutations of PI3K are observed in colorectal cancer, it would be interesting to test the efficiency of a combination treatment with gemcitabine and LY2780301 [45]. Akt and p38 MAPK signaling pathways are not only linked to oxaliplatin resistance, they were also shown to be strongly involved in the resistance to irinotecan and 5-FU in colorectal cancer [30,32,34,35].

## 5. Conclusions

In this study we have identified a treatment that is able to overcome the resistance of colorectal cancer cells to oxaliplatin by targeting a pathway that we described in our previous study. We recently showed that p38 MAPK and Src kinases play a crucial role in the resistance to oxaliplatin and that these kinases could be interesting therapeutic targets (Figure 5). Src and p38 MAPK inhibitors are currently under clinical evaluation for colorectal cancer treatment [10,11,46,47], but we are showing here that an existing chemotherapy, gemcitabine, could be used to target these kinases. By inhibiting Src, p38 MAPK, and the Akt signaling pathway, gemcitabine is able to induce apoptosis in oxaliplatin-resistant cells (Figure 5). Taken together these results clearly show that gemcitabine could be an alternative treatment for patients with oxaliplatin-resistant tumors associated with high p38 MAPK activity.

## Figures and Tables

**Figure 1 cancers-14-05894-f001:**
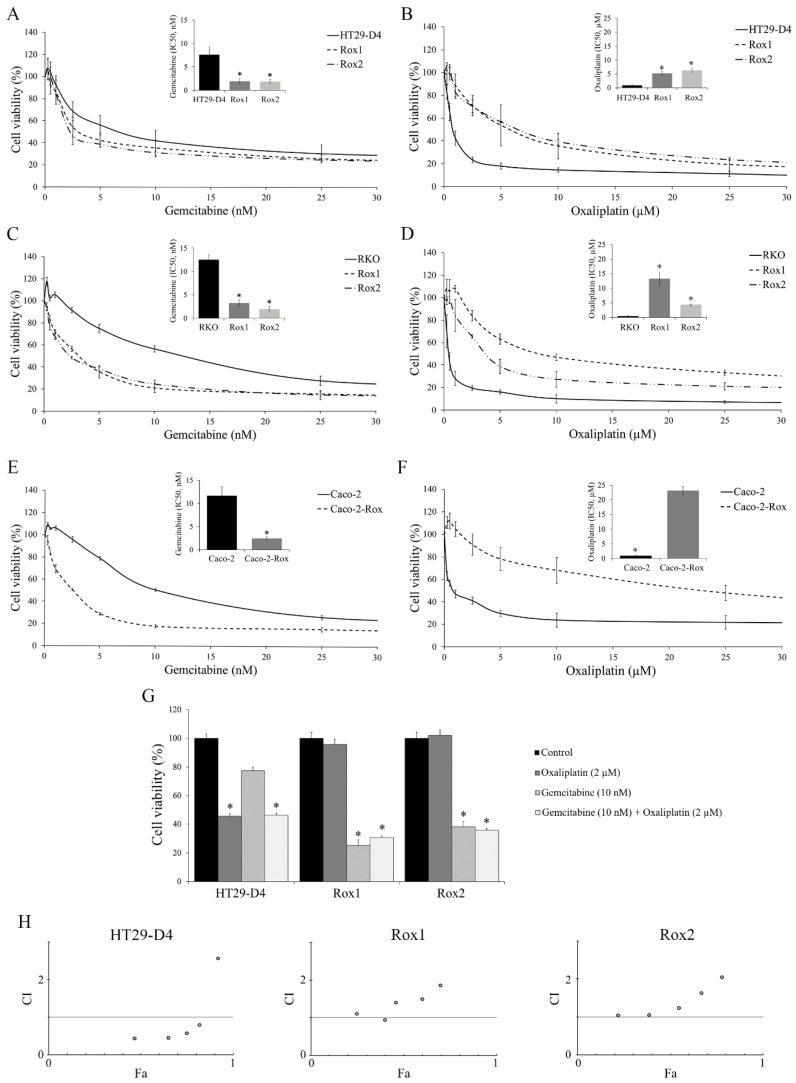
Characterization of the chemoresistance status of the cells and the effects of the gemcitabine/oxaliplatin combination. (**A**,**C**,**E**) Effects of gemcitabine on HT29-D4, RKO, and Caco-2 viability, respectively. IC_50_ values are presented in the corners. (**B**,**D**,**F**) Effects of oxaliplatin on HT29-D4, RKO, and Caco-2 viability, respectively. IC_50_ values are presented in the corners. (**G**) Effects of oxaliplatin and gemcitabine, used alone or in combination, on the viability of HT29-D4, Rox1, and Rox2 cells. (**H**) Combination status of oxaliplatin and gemcitabine in HT29-D4, Rox1, and Rox2 cells (Chou–Talalay method). Combination indexes (CI) are plotted on the *y*-axis as a function of the affected fraction (Fa; *x*-axis). CI < 1, CI = 1, and CI > 1 indicate synergism, additivity, and antagonism, respectively. Asterisks indicate statistical significance with *p* < 0.05.

**Figure 2 cancers-14-05894-f002:**
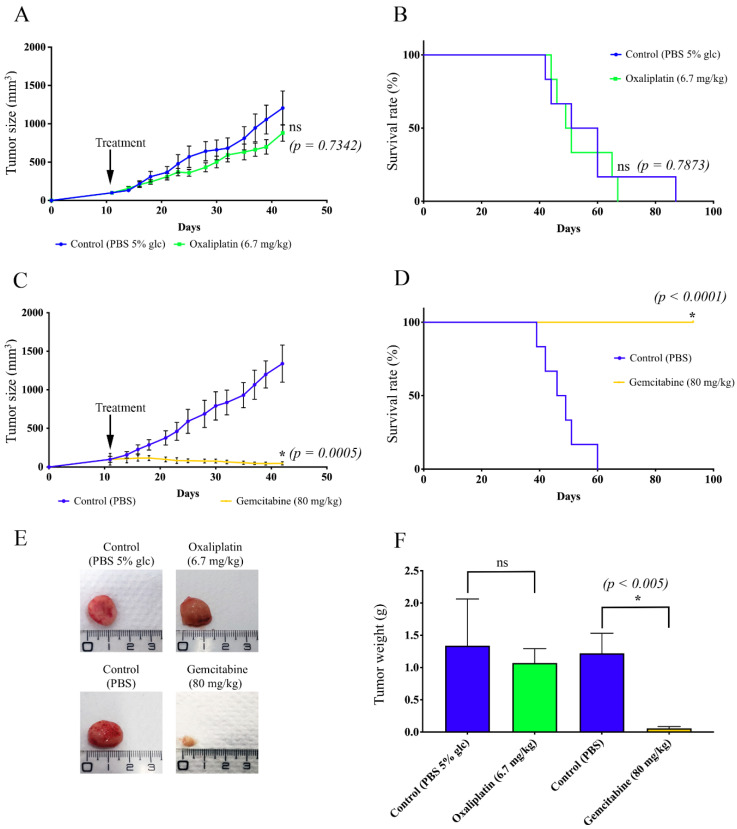
Study of the effects of gemcitabine on oxaliplatin-resistant colorectal cancer cells in vivo. Nude mice were inoculated with HT29-D4 Rox1 cells. After 11 days, the mice were treated with oxaliplatin (6.7 mg/kg), gemcitabine (80 mg/kg), PBS and 5% glucose (PBS 5% glc), or PBS. (**A**) Growth curves of the xenografted tumors for oxaliplatin and its control (PBS 5% glc). (**B**) Kaplan–Meier survival curves for oxaliplatin and its control. (**C**) Growth curves of the xenografted tumors for gemcitabine and its control (PBS). (**D**) Kaplan–Meier survival curves for gemcitabine and its control. (**E**) Representative pictures of the tumors after dissection. (**F**) Tumor weight comparison. Asterisks indicate statistical significance (*p* values are indicated on the graphs), ns: not significant.

**Figure 3 cancers-14-05894-f003:**
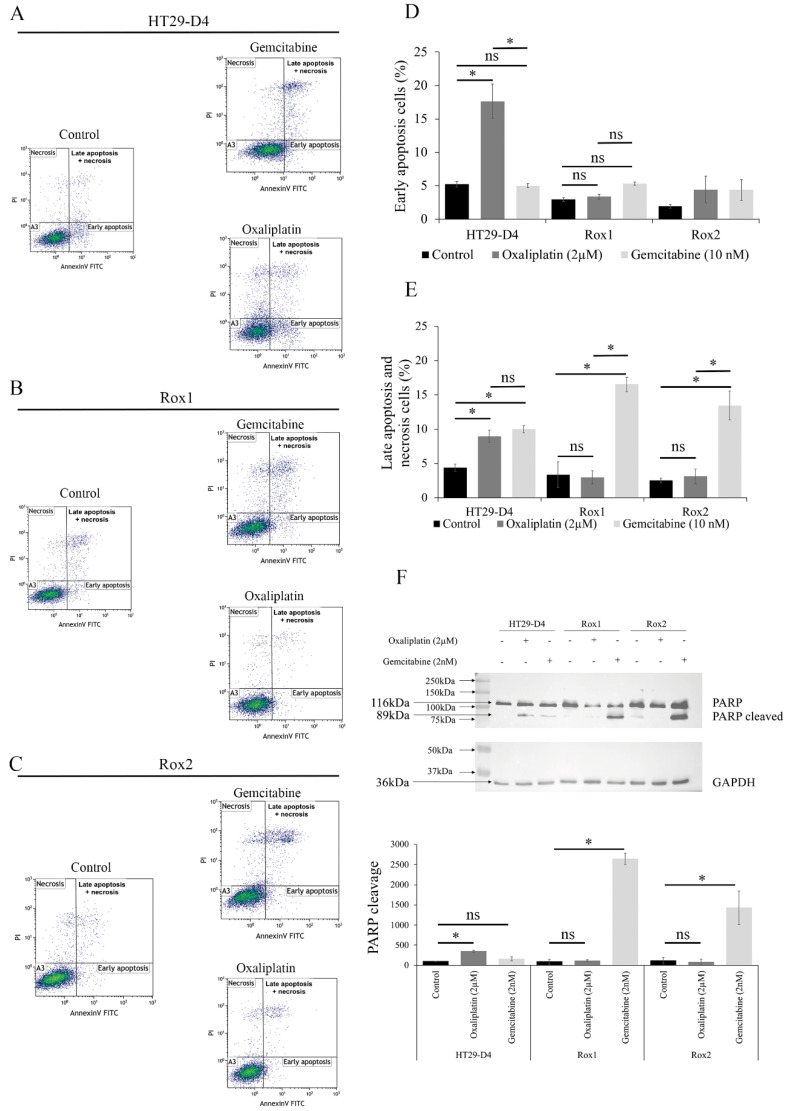
Effects of gemcitabine on the induction of apoptosis. (**A**–**C**) Representative results of the flow cytometry analysis of annexin-V-FITC and propidium iodide staining (PI) for HT29-D4, Rox1, and Rox2 cells incubated for 48 hours in the presence of oxaliplatin or gemcitabine. (**D**,**E**) Proportion of cells in early and late apoptosis (expressed as percentages). The values presented are means ± standard errors (error bars) (*n* = 3). (**F**) Study of PARP cleavage in HT29-D4, Rox1, and Rox2 cells incubated in the absence or presence of 2 µM oxaliplatin or 2 nM gemcitabine. The Western blots were quantified, and the results are expressed as ratios (cleaved PARP/total PARP; 100% for control condition). Asterisks indicate statistical significance of *p* < 0.05, ns: not significant.

**Figure 4 cancers-14-05894-f004:**
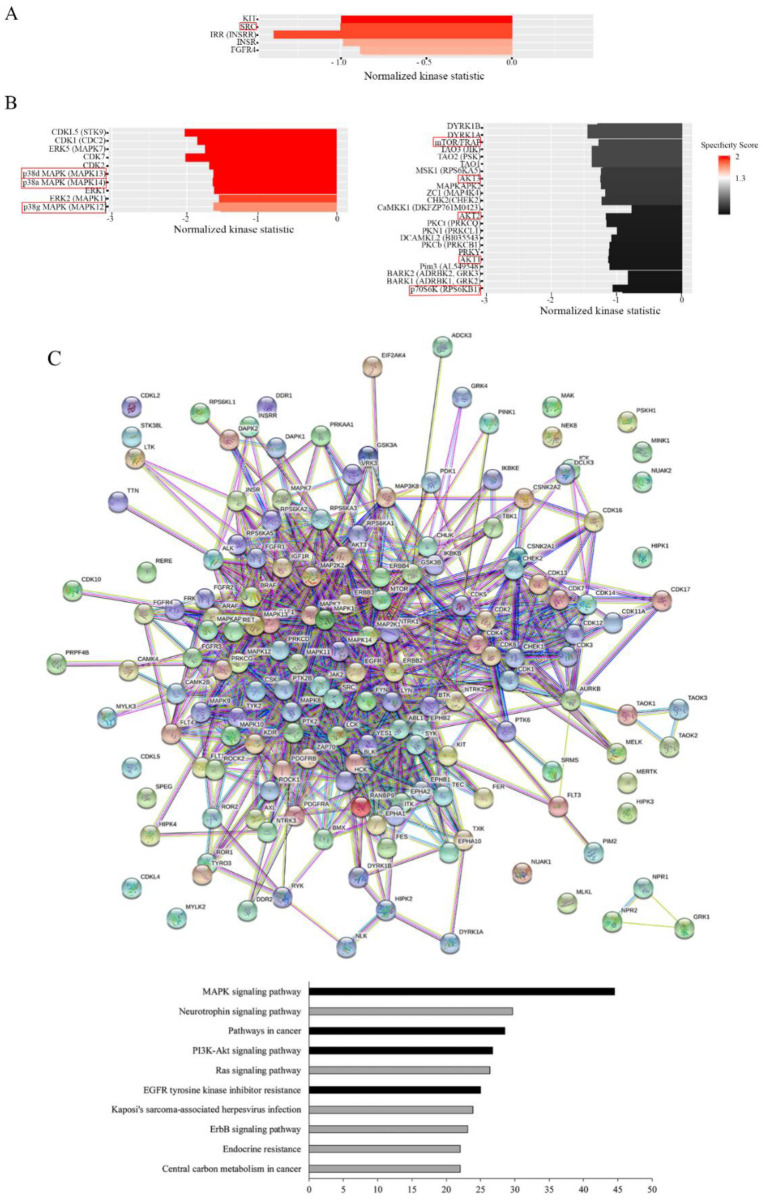
Effects of gemcitabine on kinase activity. HT29-D4 and Rox1 cells were seeded in 6-well plates and incubated in the presence of 10 nM gemcitabine for 4 hours. The cells were then lysed, and 0.5 µg of proteins were used for the PamGene kinase activity assay. The data were analyzed using the BioNavigator software to compare the kinase activities of the two cell types. The specificity scores were calculated by the PamGene BioNavigator software using six different databases. (**A**,**B**) Top kinase lists of PTK and STK analyses, respectively. A positive normalized kinase statistic value indicates a kinase activity higher in HT29-D4 cells than in Rox1 cells. (**C**) Interaction map fashioned with the String 11.0 program (https://string-db.org, accessed on 1 July 2020) showing the network nodes of all proteins analyzed with PTK and STK. The top pathway list of the interactome is also presented. Bar graphs show the -log of the false discovery rate obtained by the KEGG pathway. Black bar graphs represent the pathways of interest in cancer and drug response. (**D**–**F**) Interactomes of MAPK signaling, the PI3K-Akt signaling pathway, and EGFR tyrosine kinase inhibitor resistance, respectively. Different colored lines show the possible interactions between the nodes. The green, black, pink, red, and dark blue colors represent activation, reaction, post-translational modification, inhibition, and binding, respectively.

**Figure 5 cancers-14-05894-f005:**
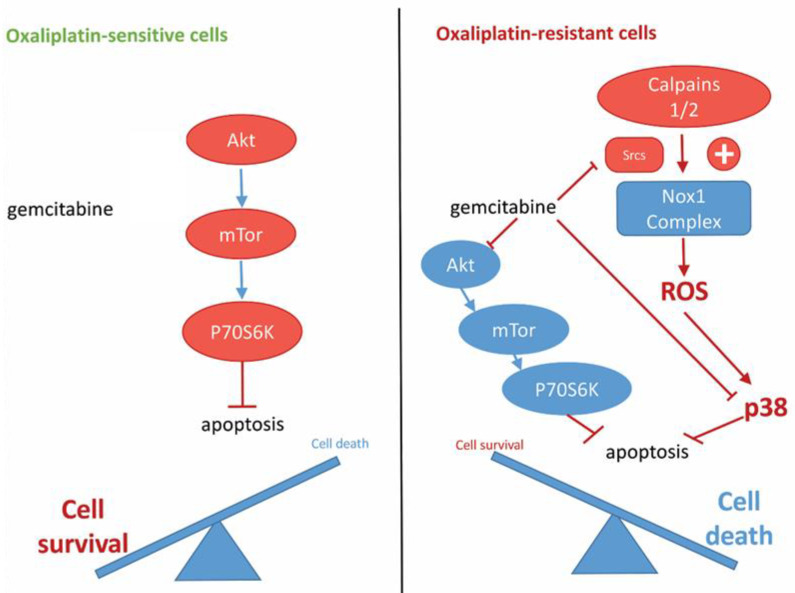
Proposed model for the mechanism of action of gemcitabine in oxaliplatin-resistant cells.

**Table 1 cancers-14-05894-t001:** Summary of the IC_50_ values calculated for the HT29-D4, RKO, and CaCo-2 cell types after 72-hour treatments with etoposide, vincristine, paclitaxel, doxorubicin, and 5-FU. The results are expressed as means ± standard deviations. * is indicative of *p* < 0.05.

Cells	Status	Etoposide (nM)	Vincristine (nM)	Paclitaxel (µM)	Doxorubicin (nM)	5-FU (µM)
HT29-D4	Sensitive	10.3 ± 1.3	3.5 ± 0.6	12.1 ± 1.8	48.0 ± 2.5	1.0 ± 0.3
	Rox1	30.8 ± 4.6 *	17.7 ± 3.5 *	34.5 ± 4.9 *	256.4 ± 8.5 *	3.4 ± 1.2 *
	Rox2	37.3 ± 2.8 *	19.6 ± 1.6 *	45.4 ± 5.4 *	300.9 ± 11.0 *	2.8 ± 0.3 *
RKO	Sensitive	9.5 ± 0.3	3.9 ± 1.5	7.6 ± 2.3	35.1 ± 4.3	1.5 ± 0.4
	Rox1	27.8 ± 3.5 *	36.3 ± 2.6 *	21.5 ± 1.9 *	250.3 ± 10.5 *	6.2 ± 1.7 *
	Rox2	22.4 ± 4.6 *	38.3 ± 3.9 *	26.8 ± 2.1 *	276.7 ± 13.4 *	7.3 ± 1.9 *
CaCo-2	Sensitive	10.1 ± 2.4	4.3 ± 1.3	8.9 ± 2.5	38.7 ± 3.7	1.3 ± 0.5
	Rox	28.3 ± 3.6 *	22.9 ± 2.6 *	28.4 ± 3.6 *	245.9 ± 20.7 *	6.8 ± 1.2 *

## Data Availability

The data and information are included in the article or Appendix A or are available from the authors upon reasonable request.

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
