# Peer review of "Gemcitabine: An Alternative Treatment for Oxaliplatin-Resistant Colorectal Cancer"

_cancers, 2022, doi:10.3390/cancers14235894_

Round 1

Reviewer 1 Report

In this study, Chocry et al. investigated what chemotherapeutic agents are effective in oxaliplatin-resistant (Rox) colon cancer cell lines (HT29-D4, RKO, Caco2). They found that out of six different chemotherapies tested – selected based on their potential to regulate the p38MAPK pathway – only gemcitabine (GEM) was effective. Furthermore, Rox cell lines had increased sensitivity to GEM compared to the oxaliplatin-sensitive parental cell lines. The authors found that GEM decreased cell viability in cultures, and inhibited growth of HT29-D4 Ros tumor xenografts. They also showed that GEM induced apoptosis in HT29-D4 Rox cell cultures, as determined by Annexin V staining and PARP cleavage. Subsequent screening of kinase activities was performed to determine which signaling pathways were altered with GEM treatment. The authors found that downregulated pathways in GEM-treated Rox cells included Src, p38MAPK, and mTOR/AKT/p70S6.

Delineating pathways involved in chemotherapy resistance and identifying ways to target them could potentially help CRC patient treatment. Overall, the manuscript is well written, and the results are interesting. However, some of the data presentation could be improved and results further validated in a revised manuscript.

Major comments:

1. The data presentation in Fig 4C-F string network diagrams is not easy to follow – they are too compacted, the text is not legible (too small), and the different colors not distinguishable from one another. The authors should find an improved or alternative way to present these data so that the reader can glean information from the diagrams.

2. The results of the PamGene kinase activity profiling screen needs to be validated by an alternative means, for example by Western blot analysis. Antibodies to investigate some these pathways were noted in the Methods, but the corresponding Western blots were not included here.

Minor comments:

1. Some of the text in the figure graphs is quite small. This pertains to multiple subfigures within in Fig 1-4, but it is particularly difficult to read Fig 1A-F inset graphs. Thus, the font size should be increased for these multiple occurrences.

2. It is also hard to quickly locate the description for each subfigure in each figure legend as written. It would be simpler for the reader if the authors used standard formatting for this, eg. (A) [Description], (B) [Description], etc.

3. The graphs in Fig 1H are scantily labeled and not self-explanatory. The figure legend for Fig 1H should include more detail on what “CI” and “Fa” are abbreviations for, and the x-axis could use some additional labeling.

4. It is not clear how the quantification of the cleaved PARP was done or what is the y-axis in the graph in Fig 3F. The authors should indicate this in the figure legend, and also provide details in the Methods. As written, the corresponding text in the Results (lines 314-320) is rather confusing.

Reviewer 2 Report

This study reports on colon cancer cell lines resistant to oxaliplatin that display cross-resistance with several other anticancer drugs but increased sensitivity to gemcitabine.

Comments and suggestions: 

P2L39: Per age-standardized incidence rates (world, both sexes, all ages, 2020), colorectal cancer was the fourth most frequent cancer diagnosis and the third most frequent cause of cancer death. The statement on P2L39 needs to be corrected accordingly. Authors should use Globocan (IARC) or other authoritative cancer epidemiology sources for support. 

P2L54-57: The discussion needs to be clarified. Resistance mechanisms are classified here as "specific to each molecule" and "common to all chemotherapies." After that, the MRP-related mechanism is mentioned, but without a clear indication of whether authors consider it to represent their first or second class of resistance mechanisms. Likewise, a specific mechanism of resistance to apoptosis is mentioned (BIRC6-related; however, it is not clear whether the authors mean to bring this as an example of "specific to each molecule" or "common to all chemotherapies." 

It should also be noted that resistance mechanisms are typically specific to groups of structurally or mechanistically related anticancer drugs rather than unique to specific drugs. 

P2L62 "These kinases, and in particular p38 α, are well known for their anti-apoptotic activity..." 

P38α can exert pro-apoptotic and anti-apoptotic effects, and these outcomes are regulated in a stimulus- and cell context-dependent manner. For this reason, the quoted statement needs to be modified accordingly because it implies only anti-apoptotic activity.  

Materials and Methods:

2.2 Reagents and antibodies: Solvents used for anticancer drugs should be reported. Authors should also state whether they had used controls with corresponding solvents for each drug exposure. 

2.3 Cytotoxicity assay: Authors should indicate whether the drugs were added to the wells immediately after plating cells or whether the cells were left to adhere and grow for some time before drug treatment. 

"The IC50 were calculated by using Chou and Talay method"... 

Chou and Talay addressed the assessment of drug combinations' effects but did not primarily define a method to determine IC50 from the dose-response data. For this reason, this section should provide more information regarding the dose-response model used for fitting experimental data to determine the IC50 values (was it a 4-parameter logistic model or another model?; Were any constraints introduced to the Hill slope or other parameters?)

2.9 Statistical analysis: Two-way ANOVA test with multiple comparisons correction is indicated, but it needs to be clarified how it has been implemented with the in vivo data. Which two nominal independent variables were assessed for their effect on in vivo data? P-values are not reported for the test of differences between groups for these two variables and their interaction, which is expected in two-way ANOVA.

P7Line270 and other instances: The animal study does not represent an actual survival study because the study endpoint is a specific tumor size. Statistics is used for time to reach this endpoint. To avoid confusion, the endpoint "survival" should be renamed to "endpoint" or "time to endpoint" as appropriate. 

Section 3.4 Fig.3: The population of cells Annexin V-/PI+ (upper left quadrant) is labeled as "Necrosis." This reviewer disagrees with considering this cell population as necrotic. Necrotic cells are typically double positive in this assay because cell membrane pores in necrotic cells allow passage of both dyes to the cell and stain PS and nuclear DNA simultaneously. The Annexin V-/PI+ cells are most likely cell debris or another artifact. Were the cells gated by FSC/SSC prior to the analysis of the stained populations to remove cell debris? This gating needs to be performed to avoid the inclusion of cell debris in the population of cells analyzed for specific staining. 

The upper-right quadrant (Annexin V+/PI+) can include late apoptotic and necrotic cells and should be labeled accordingly. 

P12L370-371: "Chemotherapy release channel" - this reviewer cannot recognize the term. Did the author mean ABC-transporters that can efflux various anticancer drugs from cells? If these transporters were suggested to play a role, this suggestion would not explain the resistance against 5-FU and oxaliplatin, which are not substrates for these transporters.

Section 3.5 How the specificity scores were calculated needs to be described. The authors indicate decreased activity of p38 MAPK in gemcitabine-treated oxaliplatin-resistant cells in comparison to the gemcitabine-treated oxaliplatin-sensitive cells. However, the results are hard to interpret because the difference between compared groups includes the effect of gemcitabine treatment and the difference between resistant and sensitive cells (comparison across a change in two variables). Therefore, it is unclear whether the reported differences in kinase activities are attributable to the effect of gemcitabine or to basal differences in kinase activities between two different cell types. A comparison between gemcitabine-treated vs. control cells for both resistant and sensitive cells would be more informative. The same comment applies to the reported inactivation of the PI3K/Akt pathway.

Figure 4C appears to include all kinases probed by the kinase activity assays regardless of assay results, and it does not seem to convey much information. This image's information value is also reduced by presenting many elements with no informative color coding. Parameters used for STRING analysis are not specified. The interactomes in Figures 4D, E, and F are not clearly described and are hard to interpret in the context of this study. Color coding is uninformative. It would be helpful to indicate which nodes actually showed different activity in the experiment and which were just added to the interactome based on their network association. Edges connecting the nodes should be clearly described (do they mean physical interaction or co-expression or co-occurrence in references?). Overall, the part discussing STRING analysis needs to be sufficiently described and, in its present form, appears to provide little, if any, information.

Figure 5: Is the figure implying that gemcitabine activates Akt in oxaliplatin-sensitive cells but inactivates Akt in oxaliplatin-resistant cells? The suggested activation of Akt by gemcitabine in sensitive cells does not seem to be supported by the data.

Discussion: The manuscript would benefit from a more detailed discussion about the somewhat surprising finding that the development of resistance to oxaliplatin in three different colon cancer cell lines led to increased activity of p38 MAPK, which also increased resistance to a mechanistically very diverse set of anticancer drugs (two inhibitors of topoisomerase II, two microtubule-targeting drugs with different modes of action and one antimetabolite/inhibitor of thymidylate synthase). Authors suggest gemcitabine can be an alternative treatment for patients not responding to oxaliplatin. However, this statement needs more justification and support for the assumption that activation of P38 MAPK and/or PI3K/Akt are rather general mechanisms involved in developing clinical resistance of colon cancers to oxaliplatin, provided that gemcitabine indeed inhibits oxaliplatin-resistant cells through inhibition of these kinase signaling. This reviewer has concerns about the generalizability of the findings presented in this manuscript. Publicly available data for 24 colon cancer cell lines (Genomic of Drug Sensitivity in Cancer, Sanger) show a positive correlation between the sensitivity of cancer cells to oxaliplatin and gemcitabine, i.e., cells less sensitive to oxaliplatin tend to be also less sensitive to gemcitabine (please, see the attached pdf file with a figure showing a strong and significant positive correlation between sensitivity of colon adenocarcinoma-derived cell lines to these two drugs). For this reason, the conclusion should demerit the implied general applicability of gemcitabine in colon cancer patients who failed to respond to oxaliplatin-based chemotherapy. Instead, the discussion and conclusion should specify which colon cancer patients who do not respond to oxaliplatin are likely to respond to gemcitabine (are there potential biomarkers of response?).

Round 2

Reviewer 2 Report

I consider the authors' responses to my suggestions and concerns adequate, and I recommend accepting this manuscript.